# Impact of HAART Therapy and HIV Infection over Fetal Growth—An Anthropometric Point of View

**DOI:** 10.3390/microorganisms10061123

**Published:** 2022-05-30

**Authors:** Daniela Roxana Matasariu, Mircea Onofriescu, Elena Mihalceanu, Carmina Mihaiela Schaas, Iuliana Elena Bujor, Alexandra Maria Tibeica, Alexandra Elena Cristofor, Alexandra Ursache

**Affiliations:** 1Department of Mother and Child’s Health—Obstetrics and Gynecology, University of Medicine and Pharmacy (Gr. T. Popa), 700115 Iasi, Romania; daniela.matasariu@umfiasi.ro (D.R.M.); mirceaonofriescu@yahoo.com (M.O.); emih2001@yahoo.com (E.M.); bujor.iuliana@gmail.com (I.E.B.); alexandra.tibeica@yahoo.com (A.M.T.); carpalecsandra@yahoo.com (A.U.); 2Department of Obstetrics and Gynecology, Cuza Voda Hospital, 700398 Iasi, Romania; michy_doctor@yahoo.com

**Keywords:** HIV, HAART therapy, birth weight, pregnancy

## Abstract

Human immunodeficiency virus (HIV) infection cannot be completely eliminated from the body because the virus integrates its genetic code into that of the host cell. The prevalence of pregnancy in women with HIV infection has increased due to the efficacy of antiretroviral therapy (ART). Placental insufficiency is associated with a reduction in blood flow and circulatory redistribution, resulting in fetal hypoxia and nutrient deprivation as a consequence of an altered placental function, and it can result in a lower birthweight. The aim of the study was to determine the combined effect of HIV infection and ART on the anthropometric parameters of infants born to HIV-positive pregnant women under ART compared to the values of these parameters in a control group of infants born to healthy mothers. There are no significant differences between the two groups in terms of gestational age at birth. We found a statistically significant lower birth weight in infants born from HIV-positive mothers under ART, with 3041 g in the control group compared to 2758 g in the group of HIV positive pregnant women (*p* < 0.01). There were statistically significant differences in all anthropometric parameters, these showing higher values in the control group (seronegative pregnant women).

## 1. Introduction

Human immunodeficiency virus (HIV) infection cannot be completely eliminated from the body because the virus integrates its genetic code into that of the host cell. After a period of time without treatment, the infection triggers a disease called AIDS, which is the acronym for acquired immunodeficiency syndrome. The percentage of HIV-positive patients is extremely variable between different regions of the globe and also between different social categories. Thus, the incidence of the disease among pregnant women is estimated to be 30–40% in South West Africa and 0.03–0.3% in the USA. There are over 18 million women of childbearing age that are infected with HIV in the world, and about 1.4 million HIV-positive women become pregnant every day. In Romania, the exact incidence is still unknown [1,2,3].

Many HIV-positive women are of childbearing age. The prevalence of pregnancy in women with HIV infection has increased due to the efficacy of antiretroviral therapy (ART). The antiretroviral therapy administered to pregnant women aims to decrease maternal viral load, thus preventing the vertical transmission of the infection. Due to the changes caused by pregnancy over the mother’s body, causing a decrease in the effectiveness of ART, the recommendation is to initiate highly active antiretroviral therapy (HAART) in all pregnant women. In pregnancy, the pharmacokinetics of antiretroviral drugs are altered, higher doses being required, due to the blood volume expansion that is characteristic for this period. Nausea and vomiting also interfere with the drug’s absorption [2,4,5,6].

The management of these cases is difficult due to long-term treatment, with many doses to be administered daily, but also due to fears related to a possible fetal harp. Other sources of non-compliance with the treatment are social and cultural stigma, side effects of therapy (nausea, vomiting, diarrhea, loss of appetite), but also financial causes. The last are related to pregnancy follow-up and transportation difficulties to centers that provide a correct and proper monitoring of pregnancy and treatment but also neonatal prophylaxis. HIV infection in pregnancy is considered to be one of the most devastating, with profound both medical and ethical implications [2,4,5,6].

Constitutionally small or small for gestational age (SGA) fetuses refers to infants born with a birth weight less than the 10th centile for gestational age, not accompanied by circulatory phenomena of blood flow redistribution (supraunitary cerebroplacental ratio) [7]. Subsequently, these fetuses present minimal risks. Fetal growth restriction (FGR) is the failure of the fetus to achieve his/her complete predetermined growth potential. These fetuses have a weight below the 3rd percentile, an abdominal circumference below the 10th percentile for the respective population and for gestational age at that time, and the Doppler evaluation reveals a circulatory redistribution. Placental insufficiency is associated with a reduction in blood flow and circulatory redistribution, resulting in fetal hypoxia and nutrient deprivation as a consequence of an altered placental function [7,8,9].

The incidence of FGR ranges between 4% and 8% in developed countries, reaching up to 25% in developing and underdeveloped countries. It is associated with increased perinatal mortality and morbidity. Morbidity and mortality rates increase dramatically and directly proportional with the decline in fetal weight, from eight times higher than in newborns without this pathology, in case the fetal weight is below the 10th percentile, up to twenty times higher in case the fetal weight is below the 3rd percentile [7,8,9,10].

## 2. Materials and Methods

This retrospective study was conducted in “Cuza-Voda” University Hospital of Obstetrics and Gynecology, Iasi, Romania, and covered 6 consecutive years, from 2013 to 2018. The study followed the changes in anthropometric parameters in infants born from HIV-positive mothers, under HAART therapy, compared to a control group of infants born from mothers without this disease, and the rate of fetal growth disorders in the group of HIV positive pregnant women compared the incidence of this pathology in all seronegative pregnant women who gave birth during this interval in our service.

Approval was obtained from the Ethics Committee of the University of Medicine and Pharmacy “Gr. T. Popa”, Iasi, Romania, and written consent was obtained from all the participants (No. 24411/11.05.2018).

As inclusion criteria, we considered: infants born to mothers known to be seropositive prior to pregnancy, currently under ART, with correct staging of the disease performed by a specialist in infectious diseases. The staging of the disease was completed by the infectious disease doctor according to 2008 CDC classification system, depending on CD4 T-lymphocyte count [11].

In terms of exclusion criteria, we mention pregnant women diagnosed with HIV infection in the moment of birth, via rapid tests, not yet confirmed by ELISA and Western blot tests. We excluded these cases because of the lack of information about the actual seropositive status, viremia, staging of the disease and treatment.

The control group included infants born from pregnant women without HIV infection with spontaneous singleton pregnancies. All cases of mothers with autoimmune diseases that required immunosuppressive treatment during pregnancy, with an active infectious disease such as rubella, toxoplasmosis, cytomegalovirus, etc., and also pregnant women with chronic renal disease, were excluded from the control group because of the possibility of impaired fetal growth affecting the results.

The study included 85 HIV-positive pregnant women who met the above-mentioned criteria and a control group of 89 healthy pregnant women. In both the control group and HIV-positive group, we included patients with a normal body mass index.

All the women were Caucasians and all of them were tested for gestational diabetes, using a 75 g glucose tolerance test, between 24 and 28 weeks of gestation. The medical records of the patients in both study groups and those of the infants were examined in order to detect particular aspects regarding anthropometric findings.

The maternal age, parity, personal medical history and obstetric history of current pregnancy of the patients in the two study groups are presented. Information about hepatitis B or C co-infection was documented.

We collected data on time of birth, type of birth (vaginal or C-section), infant gender, birth weight, gestational age, anthropometric parameters assessed at birth and infants’ developmental progress until they were discharged from hospital. We documented the Apgar score for each newborn. The score evaluates five parameters: the color, the heart rate, reflexes, muscle tone and respiration [12]. Gestational age was established using the last menstrual period and corrected, if necessary, with first and second trimester ultrasound assessments, and corroborated with immediate newborn assessment performed by neonatologists, with gestational age assessment by Ballard score. All anthropometric assessments, birth weight, height, head circumference and body mass index were performed using standard methods (Table 1).

### Statistical Analysis

The obtained data were processed using SPSS version 25.0 (IM SPSS Inc., New York, NY, USA). We applied the following: *t*-test (Independent Samples Test), Chi-Square Tests and NPar Tests—Mann–Whitney Test.

## 3. Results

This section may be divided by subheadings. It should provide a concise and precise description of the experimental results, their interpretation, as well as the experimental conclusions that can be drawn (Table 2).

### 3.1. Age, Gestational Age, Sex

The age of the patients ranged from 16 to 34 years. There were no significant differences between the two groups in terms of gestational age at birth. There were no significant differences between the two groups in terms of the distribution of the infants by sex, the sex ratio being balanced, with a *p* value of 0.051. Most cases, 45.88% (39 cases), came from Suceava County, 30.58% (26 cases) came from Neamt County and 23.52% (20 cases) came from Iasi County. Regarding the urban/rural distribution of the cases, amongst the HIV+ group, there were 45 cases in the urban area (52.94%) compared to 40 cases (47.05%) in the rural area. In the control group, there were 50 cases (56.17%) in the urban area and 39 cases (43.82%) in the rural area (Figure 1).

### 3.2. Fetal Weight

We found a statistically significant lower birth weight in infants born to HIV-positive mothers on ART, with an average of 3041 g in the control group compared to 2758 g in the group of HIV-positive pregnant women (*p* < 0.01; standard deviation for the control group was 575.407 and for the HIV-positive pregnant women group was 530.618).

Being a retrospective study, the accuracy of the diagnostic of SGA/FGR was also based on immediate newborn assessment performed by neonatologists. Thus, in 17.43% of births, the infants were SGA/FGR, both databases—maternal obstetrical diagnostic and newborn assessments performed by neonatologists—being concordant. The effective incidences of SGA/FGR in seronegative pregnant women recorded in our hospital over these 6 years were: 2.61% (151 cases) in 2013; 3.04% (172 cases) in 2014; 3.73% (222 cases) in 2015; 3.71% (236 cases) in 2016; 3.40% (210 cases) in 2017 and 3.43% (219 cases) in 2018. These values are close, without too much variability between years, the percentage remaining approximatively constant. In the HIV-positive pregnant women, the percentages of fetal growth disorder were as follows: 11.76% in 2013; 9.09% in 2014; 22.2% in 2015; 13.33% in 2016; 25% in 2017 and 13.33% in 2018. The SGA/FGR pathology in pregnant seropositive women in the C3 stage of the disease was recorded in 46.15%, one case in stage A3, 23.07% in stage C2, 15.38% in stage B2 and one case in stage C1. The distribution of SGA/FGR according to infant sex is relatively balanced. The extreme ages accounted for one case each, a woman aged 16 years old and one aged 34 years old. The mean fetal weight in stage 1 HIV-positive pregnant women was 2930 g.

### 3.3. Condition at Birth

Next, we analyzed the following parameters: Apgar score, length of stay as well as fetal anthropometric parameters at birth: fetal weight (FW), fetal height (FH), head circumference (HC) and thoracic circumference (TC). Apgar score did not differ significantly between the two groups, but there were statistically significant differences in all anthropometric parameters, these showing higher values in the control group (seronegative pregnant women). FW was 95 in the controls and 78.52 in the seropositive patients, with a *p* value of 0.03 and FH 89 cm in the control and 84 in the seropositive mothers, with a *p* value < 0.001. HC and TC had higher values in infants born from healthy mothers compared to those born to seropositive mothers on antiretroviral therapy, with a *p* value < 0.001 (Table 3).

Subsequently, we assessed some of the parameters that reflect the condition of the newborn immediately after birth. We did not find any significant differences between the two groups in terms of the incidence of respiratory distress syndrome, acute fetal distress, hypocalcemia and rate of ICU admission. We do not reach statistical significance in the case of any of these variables.

### 3.4. Co-Existing Gestational Diabetes and Hepatitis Co-Infection

All the patients were evaluated for gestational diabetes. Only one case from the HIV-positive women confirmed with gestational diabetes. The patient gave birth to a 2950 g feminine sex child thorough C-section at 39 weeks of gestation. Five patients from the seronegative control group were detected with gestational diabetes (Table 3).

Only one case from the seronegative group was tested positive for type B hepatitis. From the seropostive group, there were 16 cases with type B hepatitis, one case with type C hepatitis and one case with hepatitis B and C co-infection (Table 4).

### 3.5. Antiretroviral Treatment

All the HIV-positive pregnant women included in the study had undetectable viral loads. They were tested in the first trimester and at 35–36 weeks of gestation. Fifty-six of the patients were on Kaletra and Combivir associated treatment; 10 were on Kaletra and Isentress; 7 were on Norvir, Kivexa and Darunavir; 4 were on Kaletra Combivir and Isentress; 4 were on Combivir and Isentress; 4 were on Kaletra, Combivir and Isentress.

## 4. Discussion

Most cases of vertical infection occur in regions and countries with limited resources. Numerous studies in the literature report an increased risk of premature birth and impaired intrauterine fetal growth such as SGA or FGR in HIV-positive mothers. Newborns to HIV-positive mothers are smaller constitutionally compared to those of mothers without this disease. These fetal growth disorders caused by HIV infection result in a much higher rate of vertical transmission and also higher infant morbidity and mortality [10,13,14,15,16,17].

Newborns are classified according to their weight as SGA—small for gestational age, weight below 2500 g; AGA—appropriate for gestational age, weight between 2500 g and 3999 g; and LGA—large for gestational age, over 4000 g. SGA and LGA infants have an increased risk of mortality and morbidity, accompanied by a risk of antepartum, intrapartum and postpartum complications. Fetal head length and circumference are also important prognostic factors of perinatal mortality, reflecting intrauterine growth [18,19]. Thus, infants less than 47 cm in length and with less than 33 cm in head circumference are considered small. Very low birth weight (VLBW) is defined as a weight between 500 and 1500 g, hypotrophy is defined as a newborn less than 2500 g in weight and less than 47 cm in length [7,18,19].

The SGA/FCR ratio reported in the literature ranges from 3% to 7% of all newborns, being 6.2% in the United States. In our hospital, the SGA/FGR rate in normal pregnancies without HIV infection is within the values reported in the literature. In the seropositive pregnant women, there is an increase in the incidence of this condition, which occurs both due to the actual viral infection and as a result of antiretroviral therapy, especially due to the use of protease inhibitors or NNRTIs in treatment regimens [20,21].

No statistically significant variations in Apgar score at birth were detected. In addition, the sex distribution of newborns was balanced between the two groups, excluding possible statistical errors arising from it.

We also noticed differences in term of length of stay for these newborns. A reduced number of inpatient days was found in the control group (*p* < 0.01). It comes out that more frequently, the children born to HIV-positive mothers present with conditions and complications requiring longer stays. We aimed at determining whether the most common complications in the immediate postpartum were the cause of longer stay. No statistically significant differences between the two groups were found in terms of the incidence of respiratory distress, hypocalcemia, fetal distress or the need for hospitalization in the neonatal intensive care unit.

The co-administration of ritonavir/lopinavir as enzyme inducers causes early dysfunction of the adrenal gland, either by a direct effect on a developing structure or by an indirect ACTH effect [17,18]. Antiretrovirals cross the placental barrier in variable amounts, their effects on fetal growth and development being still under investigation [22,23,24]. It is difficult to assess the effect of antiviral drugs separately from that of HIV infection on the fetus due to the early and effective diagnostic of this disease and the extremely rare cases of HIV-positive pregnant women who are not on antiretroviral medication. In addition, the impressive results of this medication on the progression of HIV positive cases overshadow any negative consequences. Therapy is essential for both mother and fetus, dramatically reducing the vertical transmission of a condition with a major impact on the course of disease, mortality and morbidity of those affected [19,23,25]. The SMARTT study was designed to establish the effects of maternal ART on children, due to evidence in the literature regarding fetal growth impairment, development of metabolic disorders, fetal neurological and cardiac impairment [26].

The most used anthropometric parameters are weight and height. Height reflects bone growth. Increase in length/height is a much slower process that weight gain, but once a certain level is reached, it cannot be lost. Another parameter is head circumference (occipitofrontal), which is an indicator of the intracranial volume. These measurements are easy to perform, non-invasive, and the values are comparable with the results of other studies [27,28,29].

Head circumference correlates with brain volume, especially in young children. The central nervous system is one of the main targets of HIV infection. Sometimes, infection with this virus is associated with slower brain growth and development. A smaller head circumference at birth was correlated with a delay in subsequent neurocognitive development [30,31,32].

Sangeeta et al. demonstrates lower values of the anthropometric parameters in infants born from HIV-positive mothers, values directly correlated with maternal immune status, clinical stage of the disease and their weight [22,24].

Ekali et al. emphasize the negative impact on fetal growth of inflammation, immune activation and oxidative stress in exposed but uninfected newborns. The consequences could be much more devastating than they initially seemed through the phenomenon of “programming” having a long-term impact through the life of the individual [33].

Evans et al. reported a low birth weight, accompanied by smaller head length and circumference in infants born to HIV-positive mothers on ART compared to the control group. On the other hand, the body mass index does not seem to be affected. They mention a significant impact of maternal HIV status on the anthropometric parameters of newborns even in the absence of vertical transmission of the infection [30].

The possibly involved mechanisms in the occurrence of growth deficit in children exposed to maternal HIV infection, but uninfected, are:Chronic systemic inflammation that causes a decrease in the synthesis of insulin-like growth factor;Impairment of infant intestinal microbiota in the context of maternal HIV infection, with subclinical intestinal involvement and occurrence of inflammation;FGR and premature birth associating a suboptimal fetal growth both after delivery and in the first 12 months of life;Severity of the maternal disease stage;Maternal and nutritional status;ART: PI and NNRTI seem to cause premature birth and fetal growth disorders such as SGA/FGR;Socioeconomic factors such as educational level [34,35,36].

Approximately 25%of patients presented co-infection with one or both hepatitis viruses B and C [37]. Their incidence is quite high compared to the rated reported in the European literature, but it is close to that rated in African regions. The course of pregnancy and maternal and fetal prognosis are further influenced by this aspect [37,38]. Any chronic viral infection severely affects the course of pregnancy and fetal growth.

The main limitation of our study was the fact that it was a retrospective study we did not have the opportunity of detailed matching of the seropositive cases with controls.

## 5. Conclusions

Concluding, the data in the literature point out that the differences in anthropometric measurements of children born to HIV-positive but uninfected mothers are present from the time of intrauterine life.

The anthropometric parameters of the infants reflect their nutritional status. Data on fetal growth are important in assessing the prognosis of the disease, the efficacy and toxicity of ART, but also in assessing the nutritional impact of HIV infection. Growth may be one of the most sensitive indicators of disease progression in HIV-positive children. Even in children on ART, the absence of growth becomes an indicator of unfavorable prognosis. In contrast, weight gain is an important indicator of treatment efficacy. Thus, anthropometric parameters become useful in monitoring disease progression and in assessing the response to therapy. As far as we know, it is the first study conducted in Romania that evaluated the parameters of fetal growth in HIV-positive pregnant women on ART.

In recent decades, significant progress has been made with respect to the vertical transmission of HIV infection, especially through ART administered during pregnancy. Consequently, most children affected by this pandemic are those exposed to maternal infection, being born to HIV-positive mothers treated with antiviral drugs but not infected. Infants exposed to maternal HIV infection but uninfected have increased mortality and morbidity compared to non-exposed ones.

The period of intrauterine development is one of the most vulnerable stages of development, fetal growth restriction having significant consequences in fetal, neonatal, and adult life. Growth retardation is associated with unfavorable neonatal outcomes, including increased mortality, mainly due to low birth weight. The impact persists, and it is felt later in adult life. Conducting studies on the anthropometric aspects in infants born to HIV-positive mothers on ART could prove extremely useful and could contribute to the development of interventional strategies targeting this population segment.

## Figures and Tables

**Figure 1 microorganisms-10-01123-f001:**
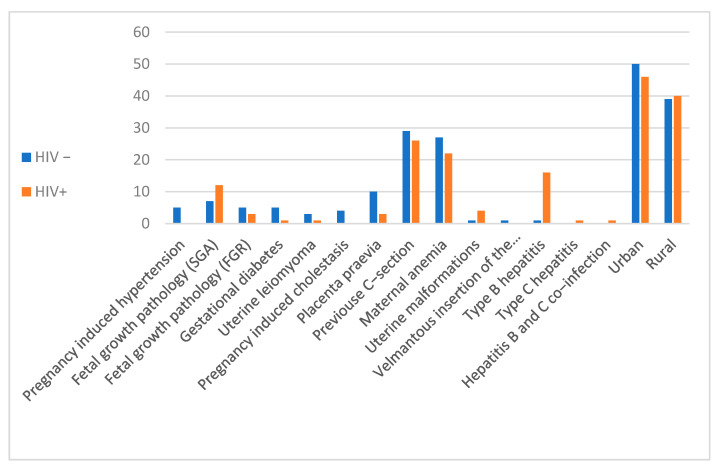
Distribution of cases by demographics and associated pathologies.

**Table 1 microorganisms-10-01123-t001:** The methods used to evaluate newborn babies.

Measured Parameter	Length	Cranial Circumference	Thoracic Circumference	Weight
Method	Infantometer	Circumference tape	Circumference tape	Baby scale—Electronic weighing scale

**Table 2 microorganisms-10-01123-t002:** The statistical distribution of the evaluated parameters.

Evaluated Parameter	Normal Distribution—Yes (Y)/No (*n*)	Statistical Significance—Yes (Y)/No (*n*)
Weight	Y	Y (*p* < 0.01)
Sex	Y	*n*
Apgar score	Y	*n*
PI (Ponderal Index)	Y	Y (0.03)
FH (fetal height)	Y	Y (*p* < 0.01)
HC (cranial circumference)	Y	Y (0.01)
TC (thoracic circumference)	Y	Y (0.01)
Number of hospitalization days	Y	Y (0.01)
Respiratory distress	*n*	*n*
Hypocalcemia	*n*	*n*
NCIU admission (Neonatal intensive care unit)	*n*	*n*
FGR/SGA/AGA/LGA (Fetal growth restriction/ Small for gestational age/ Appropriate for gestational age/ Large for gestational age)	*n*	*n* (Equal proportions between the 2 groups)

**Table 3 microorganisms-10-01123-t003:** Anthropometric parameters in infants born from HIV positive patients compared to control group.

Anthropometric Parameters	Ponderal Index	Fetal Height	Cranial Circumference	Thoracic Circumference	Apgar Score	Number of Hospitalization Days
HIV-positive women (mean)	2.3925 (SD 0.21226)	48.44 (SD 3.125)	32.554 (SD 2.0855)	31.560 (SD 2.3814)	8.23 (SD 0.812)	8.87 (SD 11.368)
Control group (mean)	2.4464 (SD 0.21226)	50.20 (SD 3.507)	33.037 (SD 2.4391)	32.365 (SD 2.1934)	8.27 (SD 1.223)	5.81 (SD 6.362)
*p*-value	0.030	<0.001	<0.001	0.009	0.177	0.002

**Table 4 microorganisms-10-01123-t004:** Associated pregnancy pathologies and statistical significance.

Associated Pregnancy Pathologies	HIV Positive	HIV Negative	*p*-Value
Pregnancy-induced hypertension	0	5.61%	0.03
Fetal growth pathology (SGA/FGR)	13.95%/3.48%	7.86%/5.61%	<0.19/0.50
Gestational diabetes	1.17%	5.61%	0.11
Uterine leiomyoma	1.17%	3.37%	0.33
Pregnancy induced cholestasis	0	4.49%	0.05
Placenta praevia	3.52%	11.23%	0.05
Previous C-section	30.58%	32.58%	0.74
Maternal anemia	25.88%	30.33%	0.49
Uterine malformations	4.70%	1.12%	0.16
Velamentous insertion of the umbilical cord	0	1.12%	0.33
Type B Hepatitis	20%	1.12%	<0.001
Type C Hepatitis	2.35%	0	0.15

## Data Availability

Publicly available datasets were analyzed in this study. This data can be found here: https://osf.io/3e7ra/?view_only=f10408f3a4404eadb5b6b8f842b8e82c (accessed on 7 May 2022).

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
