# Peer review of "Impact of HAART Therapy and HIV Infection over Fetal Growth—An Anthropometric Point of View"

_microorganisms, 2022, doi:10.3390/microorganisms10061123_

Round 1

Reviewer 1 Report

This is an important research comparing anthropometric parameters among infants born to mothers with and without HIV.

Some of the major comments that I have that the authors need to address are as follows:

> Methods: Were patients checked for gestational diabetes or pre-diabetes? This is very important as this can affect fetal birth weight. If so, this has to be clarified in the Methods and stated in the Results.

> Were co-infections with hepatitis checked in among the patients? If so, this has to be clarified in the Methods and stated in the Results.

> Results: Please provide citation in the Methods the classification system used for "C3" "C2" "C1" "A3", etc as this is different from the CDC Stage. Please include this in the Methods.

> Results: As placed in the Discussion, the type of ART may affect the birthweight. Is there data on the ART regimen of the patients? If so, the ART regimen of the patients have to be stated in the Results.

> Did all patients have undetectable viral load? If some patients had detectable viral load, a subset analysis would be crucial. That is, please analyze the data again EXCLUDING the patients with detectable viral load.

> Lines 136 to 142, the percentages can be misleading. Please provide the actual number of patients per year and the corresponding percentage (N). 

> Please provide the standard deviation of the infant weights (Line 129).  

> Please ensure that every major statement in the Introduction and Discussion have appropriate citations at the end of the sentence. For example: Line 50, Line 186 (USA data), Line 209 (ACTH), Line 234, Lines 260 to 270  

> Please revise Table 1 to show Column 1: Anthropometric Parameters, Column 2: HIV+ women, Column 3: HIV-seronegative control, Column 4: P-value. This way, the readers could easily compare the actual values of the Apgar score, Ponderal Index, etc between the two groups.  

Minor comments include:

> Abstract: Line 17: "The management of these cases is difficult due to poor compliance" is a broad generalization. Consider revising or removing.

> Abstract: Please state the actual differences in weight in the abstract (3041g vs 2758g, p<0.01).

> Line 31: After a period of time without treatment, the infection triggers...

 > Some language editing needs to be done especially in the conclusions for clarity. 

> Line 118 to 120 should be deleted. This is the instructions to authors. 

Reviewer 2 Report

This manuscript presents data from a retrospective study covering 6 years of data collected at the University hospital of Obstetrics and Gynecology at Iasi in Romania.

The study is very thorough with extensive literature review to support the findings and discussion.  

The English is good, with very few mistakes in spelling or grammar.

The scientific design and data analysis is sound.  The work is very critical and merits publication.

Generally speaking, the manuscript could do with less words, more tables, supplementary material availability including perhaps an online database with detailed data tables and statistical analyses.  The structure of the paper could be improved as outlined below, including perhaps writing a separate literature review on the topic: - 

  • Line 85: the word "correctly" should be changed to "correct" so that the sentence reads as "... with correct staging of the disease".
  • Lines 87 to 89: Why?  Explain why these exclusions were made.
  • Lines 90 to 91: Why?  Why were only singletons included
  • Lines 91 to 94: Why? Why were these criteria used to exclude mothers from the control group?
  • Were the cases and controls matched on any specific criteria and how do their demographic characteristics compare especially on factors that might affect child development parameters?  We need a table with statistics comparing key demographic and other health characteristics between controls and cases that might affect the study outcomes.
  • Line 109:  What "standard methods"? - specifics please.  Could do with naming the methods or providing a table or details in supplementary material.
  • Lines 114 to 116:  Need to specify exactly which measures were normally distributed and analysed one way versus which ones were not.  This could be within a supplementary table perhaps but we do need all the specific details of the methods.
  • Lines 118 to 120:  Please delete these manuscript instructions to authors.
  • We could do with a table of these results and some graphs.
  • Generally the results are very wordy and could do with more tabulation and graphing to make it easier to assess.  Table 1 is not sufficient for the amount of data analysed for this study.
  • Generally the discussion is too long and starts to become a summary of key literature or literature review.  It is very interesting perhaps for those who are not familiar with the subject area.  Perhaps the authors should consider writing a separate literature review on the topic given the extensive amount of literature analysed.  It is very informative to those not familiar with the field but for this specific manuscript is too much: - 
  • Lines 185 to 187:  Perhaps a supplementary table describing criteria for anthrometric scoring of SGA and FCR.
  • Line 196:  What is the agpar score?  This should be described perhaps in the methods.
  • Generally speaking, no new results should be introduces in the discussion.  Authors should think how to structure the manuscript so that there are clear hypotheses being tested and aims to test these hypotheses.  Clearly stated hypotheses and aims help structure the manuscript i.e., we were testing hypotheses 1 and 2.  We had aims 1a, b and c to address hypothesis 1 and aims 2a, and 2b to address hypothesis 2.  Structure the results according to the data that relates to each aim and thus hypothesis then go on to discuss the results as the related to each hypothesis in this same order so it easy to follow.  Conclusions should generally go on to state whether the data support the hypotheses. 
  • The discussion should discuss whether the results support the hypotheses and whether there are other studies in the literature that support or refute these findings and discuss any discrepancies.
  • Discussion should not really go into a detailed literature review or introduce new concepts that were not already outlined in the introduction.
  • Discussion should clearly state any limitations to the study - e.g. being a retrospective case-control study, with no detailed matching of cases and controls (is this the case?).  Authors should clearly state the limitations of their study and suggest how future studies could improve on this and address any shortcomings.
  • Authors should cut down the discussion and introduce all key concepts in the introduction and maintain a consistent structure throughout the manuscript.
  • We need more tabulation of the patient characteristics and results perhaps with supplementary tables.  More graphs.
  • It would be nice if the data was made available via an open access data base e.g. https://osf.io/?view_only=47cbcfd3d39b4a89a0d758a474441e38.  You can provide Excel tables as well as graph pad or statistical package analyses via this route and the publication of the database counts as a publication that can be cited, so I would encourage the authors to take the time to do this.  It will be useful to others.

Thank You.

Round 2

Reviewer 1 Report

The authors have satisfactorily addressed my initial comments. I recommend to publish the paper after minor editing (for example, Table 3: change p-values indicated as “.000” under Fetal Height and Cranial circumference into <0.001). Table 2 might also be better presented with the actual p-values to indicate statistical significance or not. There were also some minor spelling errors.

Nonetheless, this paper provides valuable information on the anthropometric parameters of children born to mothers with HIV that is worth publishing and sharing with the scientific community. This is especially important in the era of antiretroviral therapy.

Reviewer 2 Report

  • It would be nice if there were some statistics p-values to go with Table 4.  This is standard for such data.
  • I still think it would be nice to have access to the raw data files from which all the data in the manuscript were derived.  A link could be created and included with the manuscript even if it is still a work in progress.  This is important for transparency and to allow other researchers to verify and analyze the results further e.g.: - https://osf.io/.
  • I still think that some actual graphs would help break up the manuscript further as it is still very data-heavy.  I think some graphs to illustrate the main findings would be useful.
